# The Effect of Glycomacropeptide versus Amino Acids on Phenylalanine and Tyrosine Variability over 24 Hours in Children with PKU: A Randomized Controlled Trial

**DOI:** 10.3390/nu11030520

**Published:** 2019-02-28

**Authors:** Anne Daly, Sharon Evans, Satnam Chahal, Saikat Santra, Alex Pinto, Cerys Gingell, Júlio César Rocha, Francjan van Spronsen, Richard Jackson, Anita MacDonald

**Affiliations:** 1Birmingham Women’s and Children’s Hospital, Steelhouse Lane, Birmingham B4 6 NH, UK; evanss21@me.com (S.E.); satnamc@hotmail.com (S.C.); s.santra@nhs.net (S.S.); alex.pinto@nhs.net (A.P.); anita.macdonald@nhs.net (A.M.); 2Nottingham Queen’s Medical Centre, University Hospital, Derby Road, Nottingham NG7 2UH, UK; cerys.gingell@nuh.nhs.uk; 3Centro de Genética Médica, Centro Hospitalar Universitário do Porto (CHUP), 4099-028 Porto, Portugal; julio.rocha@chporto.min-saude.pt; 4Centro de Referência na área de Doenças Hereditárias do Metabolismo, Centro Hospitalar Universitário do Porto-CHUP, 4099-001 Porto, Portugal; 5Center for Health Technology and Services Research (CINTESIS), 4200-450 Porto, Portugal; 6Beatrix Children’s Hospital, University Medical Centre of Groningen, University of Groningen, Hanzeplein 1, 9713 GZ Groningen, The Netherlands; f.j.van.spronsen@umcg.nl; 7Liverpool University, Brownlow Hill, Liverpool L69 7ZX, UK; r.j.jackson@liverpool.ac.uk

**Keywords:** glycomacropeptide, phenylalanine, phenylketonuria, phenylalanine variability, amino acids, tyrosine

## Abstract

**Introduction:** In phenylketonuria (PKU), evidence suggests that casein glycomacropeptide supplemented with rate-limiting amino acids (CGMP-AA) is associated with better protein utilisation and less blood phenylalanine (Phe) variability. **Aim:** To study the impact of CGMP-AA on blood Phe variability using 3 different dietary regimens in children with PKU. **Methods:** This was a 6-week randomised controlled cross-over study comparing CGMP-AA vs. Phe-free l-amino acids (l-AA) assessing blood Phe and tyrosine (Tyr) variability over 24 h in 19 children (7 boys) with PKU, with a median age of 10 years (6–16). Subjects were randomised to 3 dietary regimens: (1) R1, CGMP-AA and usual dietary Phe (CGMP + Phe); (2) R2, CGMP-AA − Phe content of CGMP-AA from usual diet (CGMP − Phe); and (3) R3, l-AA and usual dietary Phe. Each regimen was administered for 14 days. Over the last 48 h on days 13 and 14, blood spots were collected every 4 h at 08 h, 12 h, 16 h, 20 h, 24 h, and 04 h. Isocaloric intake and the same meal plan and protein substitute dosage at standardised times were maintained when blood spots were collected. **Results:** Eighteen children completed the study. Median Phe concentrations over 24 h for each group were (range) R1, 290 (30–580), R2, 220 (10–670), R3, 165 (10–640) μmol/L. R1 vs. R2 and R1 vs. R3 *p* < 0.0001; R2 vs. R3 *p* = 0.0009. There was a significant difference in median Phe at each time point between R1 vs. R2, *p* = 0.0027 and R1 vs. R3, *p* < 0.0001, but not between any time points for R2 vs. R3. Tyr was significantly higher in both R1 and R2 [70 (20–240 μmol/L] compared to R3 [60 (10–200) μmol/L]. In children < 12 years, blood Phe remained in the target range (120–360 μmol/L), over 24 h, for 75% of the time in R1, 72% in R2 and 64% in R3; for children aged ≥ 12 years, blood Phe was in target range (120–600 μmol/L) in R1 and R2 for 100% of the time, but 64% in R3. **Conclusions:** The residual Phe in CGMP-AA increased blood Phe concentration in children. CGMP-AA appears to give less blood Phe variability compared to l-AA, but this effect may be masked by the increased blood Phe concentrations associated with its Phe contribution. Reducing dietary Phe intake to compensate for CGMP-AA Phe content may help.

## 1. Introduction

In phenylketonuria (PKU), there is an inability to metabolise dietary phenylalanine (Phe) into tyrosine (Tyr) due to a deficiency of Phe hydroxylase. It is treated by a low Phe diet, and the major source of protein equivalent is provided by a Phe-free or low Phe protein substitute, in the form of either l-amino acids (l-AA) or casein glycomacropeptide (CGMP-AA). CGMP is a 64-amino acid peptide, found in the whey residue produced as a by-product of cheese production from κ casein. This protein is naturally low in Phe as well as other indispensable, large neutral amino acids (histidine, leucine, tryptophan and Tyr). Supplementation with rate-limiting amino acids provides a suitable protein substitute, allowing CGMP to be administered as a protein replacement in PKU (CGMP-AA).

A disadvantage of CGMP-AA is that it contains residual Phe (typically 36 mg per 20 g protein equivalent). For patients prescribed 60 g/day protein equivalent from this source, it will supply 108 mg/day Phe, which may affect blood Phe control, especially when Phe intake by natural protein intake is low. Most children with classical PKU have a daily Phe intake varying from 3 to 8 g of natural protein. Studies examining the effect of residual Phe in CGMP-AA are largely adult-focused and limited by small subject numbers over short study time periods. In a single-dose cross-over intervention study, the metabolic effect of peptide-bound AA in CGMP-AA was examined in 8 older (aged 15 to 48 years) patients who consumed four different drinks containing either pure AA, CGMP-AA or a mixture of CGMP-AA and AA. After 240 min, no significant changes in blood Phe concentrations were reported, and they concluded that the small amounts of Phe in CGMP-AA did not increase blood Phe concentrations, although patients had high baseline blood Phe concentrations and were only studied for 4 h [1]. Ney et al. [2] examined the efficacy of CGMP-AA compared to l-AA in a randomised cross-over study, in which subjects took a low Phe diet with l-AA vs. CGMP-AA for three weeks. To account for the residual Phe in CGMP-AA, subjects were instructed to lower their dietary Phe intake. Blood Phe tended to increase (62 µmol/L) when subjects took CGMP-AA compared to a decrease when taking l-AA (85 µmol/L), although this difference did not reach statistical significance. The increased Phe concentration was associated with subjects not following protocol or failing to lower their dietary Phe intake when on CGMP-AA. Although Phe was not significantly increased, there is a suggestion that CGMP-AA plus usual low Phe diet influences Phe control. Van Calcar and coworkers [3] studied 11 subjects and reported lower Phe concentrations after an overnight fast using CGMP-AA compared to l-AA supplements, suggesting a slower release of AA in CGMP-AA. Other studies have also shown that protein synthesis and nitrogen retention are improved with ‘complete’ protein compared with individual l-amino acids [4,5]. However, further kinetic studies are required to examine if CGMP-AA is associated with a slower rate of amino acid absorption than conventional l-AA, particularly as CGMP-AA is supplemented with amino acids. 

It is established that protein substitutes consisting of synthetic mono amino acids have a lower biological and functional efficacy compared to intact protein sources [6,7]. After the ingestion of Phe-free l-AA there is a rapid rise followed by a rapid decline in plasma amino acid concentrations compared with whole protein. Non-physiological 24 h blood Phe variability is also associated with Phe-free l-AA supplements [8,9,10].

In PKU, the residual Phe in CGMP-AA may mask any potential benefit associated with protein assimilation and reduced blood Phe variability. So far, no studies have examined the impact of CGMP-AA on 24-h blood Phe or Tyr variability when using CGMP-AA as a protein substitute in PKU; also, no studies have compared 24-h blood Phe variability with and without adjustment for the Phe content of the CGMP-AA. 

The findings from a recent systematic review [11] suggest there is little evidence to support the impact of residual Phe in CGMP-AA on Phe control. However, studies to date have not been specific in addressing this question in children. Does residual Phe in CGMP-AA have a clinical impact on metabolic control? The aim of this randomised controlled cross-over study was to assess the blood Phe variability over 24 h in children with PKU, comparing conventional Phe-free l-amino supplements with CGMP-AA. In order to examine the impact of the Phe content of CGMP-AA on blood Phe variability we studied CGMP-AA under two conditions: a) no dietary adjustment for the Phe content of CGMP-AA, and b) dietary adjustment for the Phe content of CGMP-AA by lowering dietary Phe intake.

## 2. Materials and Methods

### 2.1. Subjects

We recruited 19 children (7 boys) diagnosed by newborn screening with PKU, with a median age of 10 years (range 6–16 years) from two treatment centres. Seventeen were European and two were of Pakistani origin. Inclusion criteria included being diagnosed by newborn screening, aged 6–16 years, not treated with sapropterin, known adherence with protein substitute, 70% of blood Phe concentrations within the Phe target range for age, for 6 months before starting the study and the ability to take both CGMP-AA and Phe-free l-AA supplements. Exclusion criteria were children taking sapropterin, those with co-morbidity, an inability to take CGMP-AA and Phe-free l-AA or poor blood Phe control failing to attain 70% of the routine blood Phe concentrations within the target range for 6 months prior to the study. Target blood Phe concentrations were according to the European PKU Guidelines for age [12]; for children < 12 years, 120–360 µmol/L, and for patients ≥ 12 years, 120–600 µmol/L.

The study was registered by the Health Research Authority and given a favourable ethical opinion by The West Midlands Research Ethical Committee. Written consent was obtained for all subjects from at least one caregiver with parental responsibility and written assent from the subjects if appropriate for their age and level of understanding.

### 2.2. CGMP-AA and Phe-Free l-AA Formulation

The CGMP-AA formula contained approximately 65% CGMP and 35% l-AA. It was supplemented with Tyr, tryptophan, leucine, arginine, methionine, cysteine and histidine, vitamins and minerals. One 35 g sachet of CGMP-AA (Vitaflo International Ltd., Liverpool, UK) contained 20 g of protein equivalent, and 36 mg of Phe, 120 kcal, 6.3 g carbohydrate and 1.6 g of fat, supplying 110 mg of DHA. Each sachet was mixed with 120 mL of water.

Conventional Phe-free l-AA, contained 100% mono amino acids, vitamins and minerals to meet dietary reference values. Children took their usual prescribed l-AA supplements taken from liquid pouches or as a powder taken in a semi-solid format. The Phe-free l-AA had a similar energy profile to CGMP-AA. The same dose of protein equivalent for each individual subject was taken throughout the study, and was administered at the same time (3 or 4 times daily) from both CGMP-AA and Phe-free l-AA (Table 1). 

### 2.3. Study Design

In a 6-week randomised controlled cross-over study, each subject was randomised to take three different regimens at home for 14 days (Figure 1):R1: Subjects took CGMP-AA only as their protein substitute and their usual prescribed Phe allowance from food (CGMP-AA + Phe);R2: Subjects took CGMP-AA only, but the amount of Phe contained in the CGMP was deducted from their dietary Phe allowance (CGMP-AA − Phe);R3: Subjects took Phe-free l-AA with their usual dietary Phe allowance (Phe-free l-AA).

During the last 48 h of each study period (days 13 and 14), four hourly Phe and Tyr blood spot tests were taken day and night (08.00 am, 12.00 noon, 16.00 pm, 20.00 pm, 24.00 midnight, and 04.00 am). Twelve blood spots were collected during each study regimen, with a total of 36 blood spots over the entire study for each subject. A mean blood value for each time point over 2 days was calculated for each subject. Using these values, a median value for the whole subject group was determined representing a 24 h profile.

All the children had their first blood spots taken at 8.00 am (baseline) and they were completed at 04.00 am on the third day.

Randomisation was generated by a block randomisation system and the random order was kept within a sealed envelope.

### 2.4. Standardisation of Meals

On day 12, 13 and 14 of each study period, subjects followed the same meal plan and took their protein substitute at the same time each day. This was to achieve isocaloric intake and minimise variability due to differences in daily routines and dietary patterns. In R2, when dietary Phe was withdrawn, to compensate for the Phe in CGMP-AA, the energy content was replaced using low-protein food sources or glucose polymer solutions. Parents/caregivers weighed and recorded all food on day 12, 13 and 14 during each study period.

### 2.5. Blood Spots for Phe and Tyr

Throughout the study, trained caregivers collected four hourly blood spots at home. Blood spots for Phe and Tyr were collected on filter cards, Perkin Elmer 226 (UK Standard NBS, Public Health England, London, UK). All parents/caregivers had received previous training on blood spot collection and their technique was reviewed prior to study commencement. Blood spots were returned to the laboratory at Birmingham Women’s and Children’s Hospital. All the cards had a standard thickness and the blood Phe and Tyr concentrations were calculated on a 3.2 mm punch by MS/MS tandem mass spectrometry.

All subjects received careful supervision on blood collection days with daily home visits from the lead researcher to ensure blood spots were taken at correct times and were of adequate size. Additional support was provided by telephone calls made by the lead researcher throughout the evening and night-time to ensure all blood spots were collected.

### 2.6. Statistics

Sample size calculations determined that 18 subjects were required to demonstrate a difference of 30 μmol/L variation in Phe over 48 h based on a two-sided alpha level 0.05, a power of 80% and a standard deviation of 40 μmol/L.

Data are summarised as median (range) for continuous data and frequencies of counts and percentages for categorical data. Longitudinal regression techniques were used to evaluate the change in Phe and Tyr over time. Models were constructed which included time as a categorical factor and the randomisation identifier. Patient identifiers are included as a random effect to evaluate both between and within levels of variability. Further models were constructed which included time as a continuous covariate to give the average change in Phe levels over the study period. Of secondary interest was the stability in Phe levels, and this was assessed by measuring the difference between the Phe levels at each time point and the daily median.

## 3. Results

Regarding subject withdrawal, one girl (aged 12 years) withdrew from the study on day 1 of the first regimen after she decided not to continue with the blood spot sampling.

A total of 18 children with PKU completed the study. The median natural protein intake from food was 5 g/day (range 3–30 g). A median daily dose of 60 g (40–80 g) of protein equivalent from protein substitute was consumed between three evenly divided daytime doses. The median intake of Phe supplied by the CGMP-AA was 108 mg/day (range 70–140).

At enrolment, 4 subjects had powdered Phe-free l-AA, [XP Maxamum (Nutricia Ltd., Trowbridge, UK), *n* = 1; PKU First spoon (Nutricia Ltd.) *n* = 2; PKU Express (Vitaflo International Ltd., Liverpool, UK), *n* = 1], and 15 subjects had Phe-free liquid l-AA pouches [LQ Lophlex (Nutricia Ltd.) *n* = 2, PKU Cooler (Vitaflo International Ltd.) *n* = 13]. When taking the CGMP-AA formula, all children mixed this with water.

Regarding the comparison between regimens 1, 2 and 3 for median Phe concentrations over 24 h (Table 2), there was a difference in blood Phe variability depending on the regimen that was followed. The median Phe was 290 (range 30–580 μmol/L) in R1, 220, (range 10–670 μmol/L) in R2 and 165 (range 10–640 μmol/L) in R3. There was a significant difference between the three regimens, [R1 vs. R2 *p* < 0.0001, R1 vs. R3 *p* < 0.0001, R2 vs. R3 *p* = 0.0009]. All median Phe levels were within the European PKU target guidelines (12).

Regarding the comparison between regimens 1, 2 and 3 for median Phe concentrations for each time point (Table 3, Figure 2), there was a significant difference in median Phe at each time point (i.e., 08.00 h, 12.00 h, 16.00 h, 20.00 h, 24.00 h, and 04.00 h) between R1 vs. R2, [median value *p* = 0.0027] and R1 vs. R3 [*p* < 0.0001]. No significant difference was observed between any of the time points for R2 vs. R3. The Phe concentrations were consistently higher at all time points in R1 and lowest in R3.

Regarding the comparison of the variation in Phe concentrations between regimens 1, 2 and 3 during each 8 hourly time period (08.00 h–16.00 h), (16.00 h to 24.00 h), (24.00 h to 08.00 h) (Table 4, Figure 3), measuring the Phe over 8 hourly time periods showed there was less variation with R1 and R2 compared with R3. In all three regimens, the largest change in blood Phe occurred overnight between 24.00 h and 08.00 h, R1: +35 μmol/L, R2: +20 μmol/L, R3: +70 μmol/L. The largest change in Phe at all three measured 8 hourly intervals was in R3. In all three regimens, Phe decreased between 08.00–16.00 h and from16.00 h-24.00 h, the least change was in R1, then R2 and the greatest change always in R3.

For subjects up to 12 years of age, the European blood Phe target range was maintained at 75% at each time point over 24 h in R1, 72% in R2 and 64% in R3. For subjects ≥ 12 years (*n* = 3) blood Phe target range was maintained 100% of the time in R1 and R2 and 64% in R3.

### Median Phe Concentrations for Each Subject Compared with Recommended Target Reference Ranges for R1, R2 and R3 

Comparing Phe concentrations with European PKU age related guidelines for children <12 years (120–360 μmol/L), the number of subjects with concentrations < 120 μmol/L was lowest in R1 (7%), then R2 (22%) and highest in R3 (35%). Phe concentrations > 360 μmol/L were highest in R1 (18%), then R2 (7%) and no concentrations > 360 μmol/L were recorded in R3. For older children ≥12 years (120–600 μmol/L), in R1 and R2 all Phe concentrations were within reference range, but in R3, one subject had concentrations < 120 μmol/L, and one had concentrations > 600 μmol/L (Figure 4).

For children < 12 years, the mean percentage of time over 24 h that Phe concentrations were < 120 μmol/L was 7% (SD 6) for R1 (*n* = 4), 12% (SD 8) for R2 (*n* = 7), and 15% (SD 9) for R3 (*n* = 9). The mean percentage time > 360 μmol/L was 16% (SD 4) in R1 (*n* = 4), and 25% in R2 (*n* = 1). No member of the R3 group had levels > 360 μmol/L.

In the older age group of ≥ 12 years, over a 24 h period, R1 and R2 had no Phe concentrations outside the reference range. In R3, the percentage time over 24 h with Phe concentrations < 120 μmol/L was 25% in one child, and in another child, Phe concentrations were > 600 μmol/L for 4% of the time.

Regarding the comparison between regimens 1, 2 and 3 for median Tyr concentrations over 24 h (Table 2), the median Tyr concentrations were highest in R1 and R2 (Tyr 70, range 20–240 μmol/L) and lowest in R3 (60, range 10–200 μmol/L). When median Tyr concentrations were compared over 24 h, there was a significant difference between regimens R1 vs. R3 (*p* = 0.0028) and R2 vs. R3 (*p* = 0.0016). There was no difference between R1 vs. R2 (*p* = 0.96).

## 4. Discussion

In PKU, this is the first randomised controlled study to review the 24-h blood Phe and Tyr variability using CGMP-AA compared with conventional l-AA. The impact of the residual Phe content of CGMP-AA on 24-h blood Phe and Tyr profiles under controlled conditions was also studied. When comparing each of the three regimens for the percentage of Phe concentrations within the target range for the respective ages, it was clear that the CGMP-AA minus the dietary Phe gave the most consistent results. However, this study clearly demonstrated that the Phe supplied by CGMP-AA adversely affected blood Phe control in children. In the R1 group, 18% had Phe concentrations greater than the reference range compared to none in the R3 group, suggesting that residual Phe in CGMP impacts metabolic control. However, the median Phe concentrations remained within the current targets for all three test regimens [13].

It has been well established that timing of protein substitution affects diurnal Phe variation [8]. In a study in which 16 children with PKU received a Phe-free AA every 4 h during the day and night, the median differences in blood Phe concentrations were 40 μmol/L per day. When the protein substitute was given in 3 divided doses over 10 h, the median difference in blood Phe was 140 μmol/L per day. In the current study, the protein substitute was given three times a day over the course of 12 h. The median difference in Phe, measured between the highest and lowest values, over 24 h were lowest in R1 at 35 μmol/L, followed by R2 at 45 µmol/L, and highest in R3 at 95 µmol/L. This may suggest that CGMP-AA has some influence on Phe stability compared to the l-AA group.

There is evidence to suggest that the digestion, absorption and metabolism of CGMP-AA and l-AA may vary. Amino acid availability depends directly on the quantity and quality of the dietary source of nitrogen. Protein utilization is affected by factors such as the rate of protein digestion, amino acid absorption, together with the presence and amount of other essential and non-essential amino acids and macronutrients [7,14]. The speed of protein digestion and amino acid absorption from the gut has a major effect on whole body protein anabolism. Studies measuring protein utilization have shown that a ‘slowly’ digested protein, e.g., casein, is better utilized compared with a ‘fast’ one such as milk-soluble protein isolate [15,16]. The amino acid profile of dietary proteins may also increase the secretion of insulin, which has a positive correlation with plasma leucine, phenylalanine and arginine concentrations [17,18]. Postprandial protein metabolism is a complex biochemical process influenced by the type of protein ingested, macronutrients, hormonal responses and timing of dietary intake. An even delivery of Phe into the systemic system may enable a steady state of protein utilization improving absorption kinetics increasing whole body protein balance and skeletal muscle protein synthesis.

Some studies [19,20,21,22] suggest that fluctuations in blood Phe concentrations may influence intellectual outcome, although the exact mechanism of how is unknown. In non-PKU adults, Phe fluctuates by no more than 50% over 24 h. It has been shown that the daily fluctuations in plasma-free amino acid levels are significantly affected by dietary conditions, with more frequent feeding leading to less fluctuation [23,24]. The stabilization of blood Phe concentrations would lead to less diurnal fluctuation, and this may have a positive benefit neurologically and physiologically. However, there is no agreed definition of Phe fluctuation. A review of the literature suggests that it is the cumulative effect of day-to-day fluctuations over months and years that may affect cognition impacting on intelligence quotients [25,26]. A recent short study over 26 weeks by Feldmann concluded that Phe fluctuations were negatively correlated with IQ in children with mild PKU [27]. Although no study has definitively shown that fluctuation in Phe affects neurocognitive outcome, evidence suggests there is a correlation, and therefore the stabilization of Phe concentrations should be the aim of treatment. In R2 and R3 with controlled Phe intake, the only influence on outcomes is the type of protein substitute consumed, and the evidence in this study suggests that CGMP-AA may have a positive physiological impact on reducing variability.

Tyr concentrations were highest for all three regimens at 12.00 midday and progressively declined over 24 h, with the lowest concentrations at 08.00 am. Tyr concentrations showed less stability than Phe. Only R1 showed a steady fall over 24 h. Tyr concentrations in R2 and R3, showed greater variations (rising and falling) within the 24 h period. The impact of this pattern has not been studied in detail. Tyr is an important functional amino acid; as a large neutral amino acid, it competes at the blood–brain barrier with Phe, and it is closely associated with the monoamine neurotransmitter dopamine. There is also some suggestion that Tyr from l-AA protein substitutes are less bioavailable because of altered gut bacteria compared to CGMP-AA protein substitutes [28]. More work is needed to explore the association of Tyr and bioavailability on long-term neurocognitive outcomes using CGMP-AA and l-AA protein substitutes.

There are limitations to this study; for instance, we only enrolled a limited number of children. However, because the study design was a randomised cross-over study with each child acting as their own control, it increased the power of the study. We have not accounted for energy expenditure over the three days, which could have influenced our results, although the children were at school for some of the study period and the rest of the time was closely supervised. Although protein substitute was given at the same time point each day for each child, this was not a standardised time for the group of children, but they took it according to their individual, usual daily routine.

## 5. Conclusions

This study clearly demonstrates that the residual Phe in CGMP-AA, when used as the only source of protein substitute in children, has a significant impact on blood Phe control. In its present format, the formulation of CGMP-AA should be used with caution in children < 12 years of age, as there is a risk of Phe concentrations increasing beyond target recommendations. However, the glycomacropeptide structure appears to lead to the stabilization of Phe concentrations with less fluctuation over a 24-h period compared to conventional amino acid formula. Further studies and research are necessary to refine the CGMP-AA product providing a lower Phe or Phe-free CGMP-AA, allowing greater use of glycomacopeptide, and possibly leading to better stabilization of Phe concentrations over the day.

## Figures and Tables

**Figure 1 nutrients-11-00520-f001:**
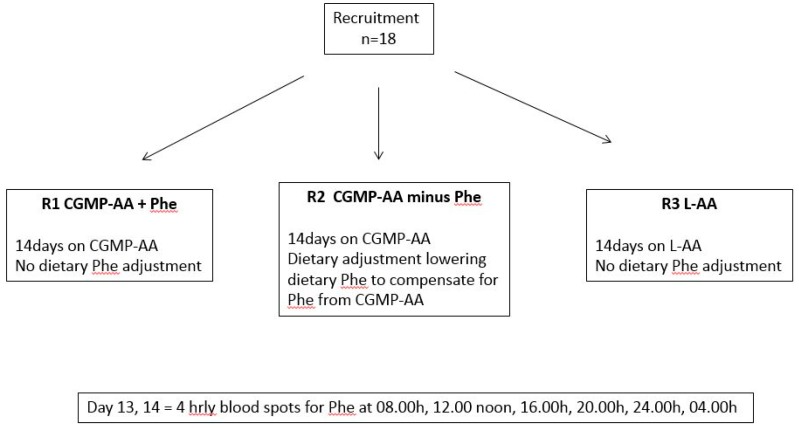
Diagram showing overview of study methodology. Each subject was randomised to R1, R2 and R3 for 14 days, on day 13 and 14, 4 hourly blood spots were collected for Phe and Tyr concentrations.

**Figure 2 nutrients-11-00520-f002:**
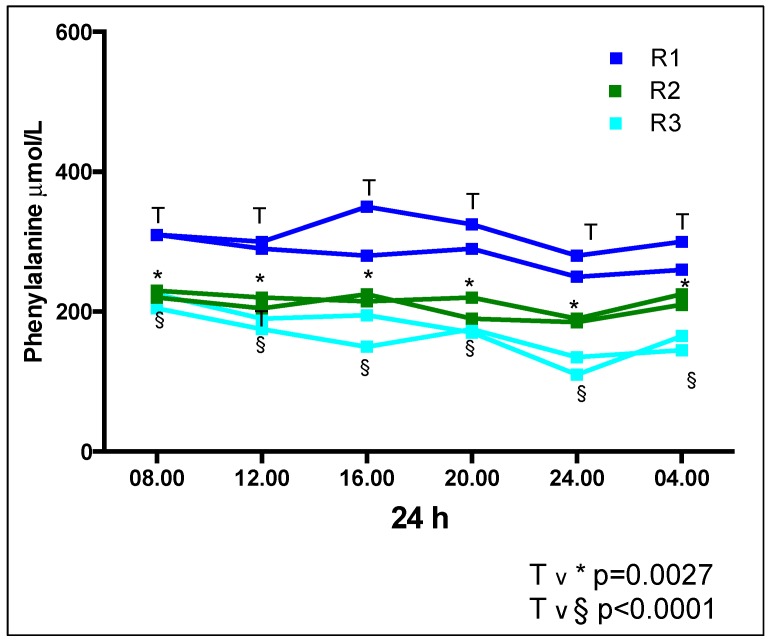
Comparison of median Phe concentrations (μmol/L) at each time point for regimens 1, 2 and 3.

**Figure 3 nutrients-11-00520-f003:**
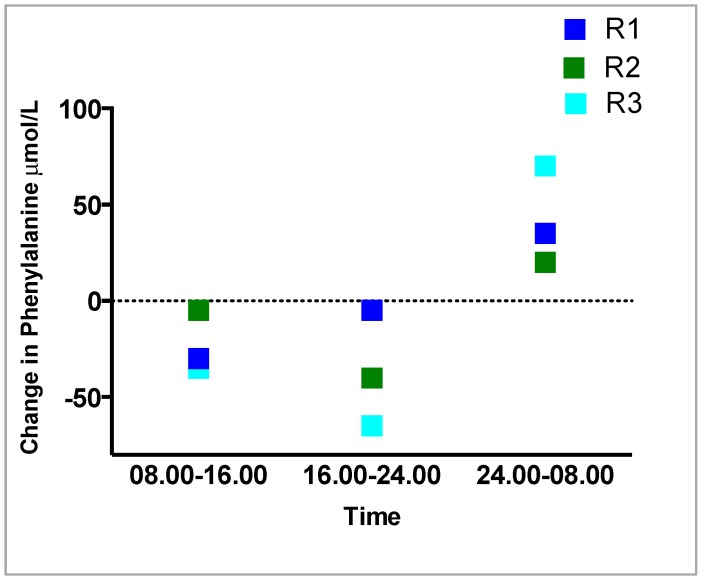
Comparison in the variation in Phe (μmol/L) concentrations between regimens 1, 2 and 3 over 8 hourly time periods.

**Figure 4 nutrients-11-00520-f004:**
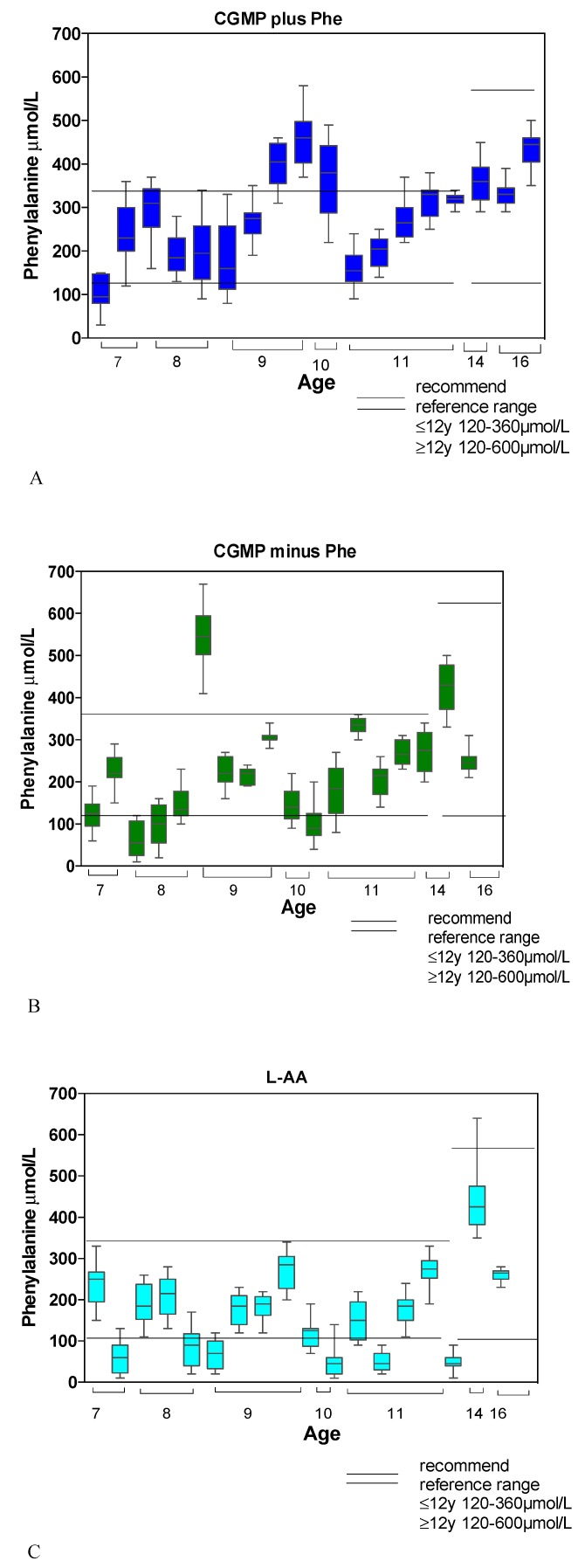
(**A**) Median Phe (μmol/L) concentrations and ranges for each subject in R1 (CGMP + Phe) compared with recommended age target ranges (target Phe range 120 to 360 µmol/L aged < 12 years; 120 to 600 µmol/L aged ≥ 12 years). (**B**) Median Phe (μmol/L) concentrations and ranges for each subject in R2 (CGMP-Phe) compared with recommended age target ranges (target Phe range 120 to 360 µmol/L aged < 12 years; 120 to 600 µmol/L aged ≥ 12 years). (**C**) Median Phe (μmol/L) concentrations and ranges for each subject in R3 (L-AA) compared with recommended age target ranges (target Phe range 120 to 360 µmol/L aged < 12 years; 120 to 600 µmol/L aged ≥ 12 years).

**Table 1 nutrients-11-00520-t001:** Nutritional composition of casein glycomacropeptide supplemented with rate-limiting amino acids (CGMP-AA) compared with the conventional Phe-free l-AAs used in the study.

		CGMP-AA 20 g PE	l-AA Protein Substitutes 20 g PE				
Nutrients	Units	CGMP-AA 1 sachet 35 g Powder Flavoured	PKU Cooler 20 1 pouch = 174 mL	LQ Lophlex 20 1 pouch = 125 mL	XP Maxamum 1 × 50 g sachet Powder Unflavoured	First spoon 50 g powder Unflavoured	PKU Express 1 × 34 g sachet Powder Flavoured
Calories	Kcal	120	130	120	149	164	101
Protein equivalent	G	20	20	20	19.5	20	20
Total carbohydrate	G	6.3	8.9	8.8	17	19.2	4.7
Sugars	G	2.2	5.9	8.8	1.6	12.8	0.3
Total fat	G	1.6	1.6	0.44	<0.25	0.6	0.07
DHA	Mg	110	134	150	0	104	0
AA	Mg	-	-	-	-	-	-
Salt	G	0.55	0.26	0	0	0	0.43
Vitamin A	µg RE	259	261	285	355	600	283
Vitamin D	µg	5.0	10	8	3.9	13.2	4.5
Vitamin E	mg αTE	5.30	5.2	3.2	2.6	7.1	5.3
Vitamin C	Mg	26	37	17.8	90	75	36.7
Vitamin K	µg	23	24	24.9	35	57	34
Thiamine	Mg	0.60	0.7	0.43	0.7	0.75	0.68
Riboflavin	Mg	0.60	0.77	0.5	0.7	0.75	0.78
Niacin	Mg	3.20	3.5	7.1	6.8	3.4	8.4
Vitamin B6	Mg	0.60	0.87	0.58	1.1	0.75	1
Folic Acid	µg	102	101	120	250	84	136
Vitamin B12	µg	1.6	1.6	1.8	1.8	1.9	1.6
Biotin	µg	13	13	53.4	70	27.8	63.9
Pantothenic acid	Mg	2.0	1.9	1.8	2.5	4.2	2.7
Choline	Mg	204	200	153	161	139	204
Sodium	Mmol	12	4.5	<10	12.1	9.1	7.5
Potassium	Mmol	5.9	6.1	<25	9	13.5	8
Chloride	Mmol	0.20	3.9	<25	7.9	9	6.9
Calcium	Mg	399	400	356	335	626	407
Phosphorus	Mmol	13	11	8.9	10.8	14.8	11.4
Magnesium	Mg	115	110	107	143	88	128
Iron	Mg	7.4	7.3	5.3	11.8	12.4	7.3
Copper	Mg	0.60	0.73	0.53	0.7	0.63	0.75
Zinc	Mg	7.4	5.6	3.9	6.8	8.8	7.3
Manganese	Mg	0.40	0.5	0.53	1.1	0.65	1.1
Iodine	µg	84	85	58.4	53.3	127	85.7
Molybdenum	µg	20	23	25	53.5	18.4	49
Selenium	µg	30	26	26.8	25	23.8	29.9
Chromium	µg	12	14	10.6	25	21	29.9
Amino acid profile							
l-Alanine	G	0.83	0.92	1.16	0.85	0.95	0.87
l-Arginine	G	0.96	1.5	2	1.6	1.65	1.41
l-Aspartic acid	G	1.31	2.37	1.75	1.5	1.55	2.23
l-Cystine	G	0.24	0.61	0.51	0.6	0.62	0.57
l-Glutamine	G	2.70		0	2.6	0	1.73
Glycine	G	0.71	2.35	1.88	1.5	1.47	2.22
l-Histidine	G	0.70	0.92	0.79	0.9	0.95	0.87
l-Isoleucine	G	1.42	1.62	1.24	1.4	1.47	1.52
l-Leucine	G	3.02	2.54	2.13	2.4	2.48	2.39
l-Lysine	G	0.95	1.67	1.63	1.9	1.71	1.58
l-Methionine	G	0.28	0.45	0.34	0.4	0.39	0.42
l-Phenylalanine	G	0.036	-	0	0	0	0
l-Proline	G	1.60	1.69	2	1.7	1.79	1.59
l-Serine	G	1.01	1.04	1.09	1.1	1.09	0.99
l-Threonine	G	2.29	1.62	1.04	1.2	1.23	1.54
l-Tryptophan	G	0.40	0.5	0.41	0.45	0.48	0.48
l-Tyrosine	G	2.25	2.38	1.88	2.1	2.2	2.24
l-Valine	G	1.14	1.86	1.38	1.6	1.58	1.76

**Abbreviations:** PE: Protein equivalent; Phe: Phenylalanine; CGMP-AA: casein glycomacropeptide supplemented with limiting amino acids; l-AA: l-amino acid (protein substitute based on 20 g PE); DHA: Docosahexaenoic acid; AA: Arachidonic acid.

**Table 2 nutrients-11-00520-t002:** Comparison of median Phe (μmol/L) and Tyr (μmol/L) concentrations (range) over 24 h for regimens 1, 2 and 3.

	R1	R2	R3
Phe μmol/L	290 (30–580) *	220 (10–670) ^T^	165 (10–640) ^TT^
Tyr μmol/L	70 (20–240) **	70 (20–220) ^§^	60 (10–200) ^§§^

Wilcoxon test Phe * v^T^, * v ^TT^
*p* < 0.0001, ^T^ v^TT^
*p* = 0.009. Wilcoxon test Tyr ****** v ^§§^, ^§^ v ^§§^, *p* = 0.002. Subtext symbols *^,TT, §,§§^ show comparisons between the groups R1, R2 and R3.

**Table 3 nutrients-11-00520-t003:** Comparison of median Phe (μmol/L) and Tyr (μmol/L) concentrations (range) at each time point for regimens 1, 2 and 3.

Time	R1		R2		R3	
	Phe μmol/L	Tyr μmol/L	Phe μmol/L	Tyr μmol/L	Phe μmol/L	Tyr μmol/L
08.00	310 (150–490) ^T^	40 (30–120)	230 (115–560) ^1^	50 (30–120)	220 (80–440) ^TT^	40 (20–200)
12.00	290 (80–450) ^T^	90 (40–150)	220 (40–600) ^2^	90 (20–170)	185 (30–480) ^TT^	80 (20–200)
16.00	280 (80–500) ^T^	90 (30–240) ^§^	225 (70–670) ^3^	90 (30–220) **	190 (20–640) ^TT^	70 (30–160) ^§§^**
20.00	300 (80–580) ^T^	80 (20–190)	210 (50–570) ^4^	75 (20–150)	170 (20–480) ^TT^	80 (10–180)
24.00	275 (30–500) ^T^	70 (30–150)	185 (10–490) ^5^	85 (30–180) ***	125 (10–440) ^TT^	60 (30–190) ***
04.00	275 (50–550) ^T^	60 (30–110) ^§^	210 (70–610) ^6^	60 (30–280) ****	150 (40–440) ^TT^	50 (30–100) ^§§^****

Wilcoxon ^T^v^TT^ p < 0.0001, ^T^v^1^
*p* = 0.0015, ^T^v^2^
*p* = 0.0028, ^T^v^3^
*p* = 0.0016, ^T^v^4^
*p* = 0.0039, ^T^v^5^
*p* = 0.0552, ^T^v^6^
*p* = 0.0073, Wilcoxon test ^§^ v ^§§^
*p* = 0.03, ** v** *p* = 0.01, *** v*** *p* = 0.02, **** v**** *p* = 0.007. Subtext symbols e.g., ^T,TT,^ **^,^***^,^**** show comparisons between the groups R1, R2 and R3.

**Table 4 nutrients-11-00520-t004:** Comparison of the variation in Phe (μmol/L) concentrations between regimens 1, 2 and 3 over 8 hourly time periods.

Time	R1 Phe μmol/L	R2 Phe μmol/L	R3 Phe μmol/L
08.00–16.00	−30	−5	−35
16.00–24.00	−5	−40	−65
24.00–08.00	+35	+20	+70

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
