# Peer review of "The Effect of Glycomacropeptide versus Amino Acids on Phenylalanine and Tyrosine Variability over 24 Hours in Children with PKU: A Randomized Controlled Trial"

_nutrients, 2019, doi:10.3390/nu11030520_

Round 1
Reviewer 1 Report
In the title PKU or phenylketonuria should be spelled out and abbreviated rather just PKU. second line phenylalnanine should be phenylalanine.
I should think that the best treatment would be reduced phenylalanine consumption to a minimum and attempt to stabilize phenylalanine concentration kinetics over an entire day. It appears that authors have accomplished the latter but not necessarily the former.
There have been two recent drugs approved in the USA -Biomarin's Kuvan which apparently binds to and enhances phenylalanine hydroxylase and Pegvaliase which is an enzyme substitute.
Maybe combining one or both of these approaches with the the best of the authors approach would be most effective.
in line 332 of the conclusions glycomacopeptide needs an r after the c =glycomacropeptide.
Author Response
Dear Reviewer
I have submitted a file outlining the changes made.
Thank you for taking the time to review this paper.

Reviewer 2 Report
Overall the authors present an interesting study.
What I believe should have more attention is what they think about the higher Phe values (above advised range for age) in the CGMP groups. In the discussion they are very positive about the results, but I would like them to also be more critical about this finding. Maybe they have suggestions about how to handle this problem?
- Spelling: the r is missing in the word children.
- A more concise title would be easier to read.
Abstract
Methods: timing of blood draw: 04h? Do you mean 4 am?
Introduction
- The aim of the study is clearly stated, but the clinical significance for doing the study is less clear. Please elaborate: is your hypothesis that CGMP-AA has benefits over the regular AA? Please keep in mind that the CGMP-AA are also supplemented with loose AA and minerals.
Material and methods
- Line 116: please start table on a new page.
- The CGMP-AA seems to contain the double amount of fats compared to L-AA. On the other hand, L-AA contains 3x the amount of sugar when compared to CGMP-AA. Do you think this should be taken in account when treating patients? Keep in mind that obesity seems to be a growing problem, also for PKU patients.
- Is the table representative for all used L-AA in your patients? As I recall, all supplements have different intrinsic values of nutrients, based on what the fabricating company has put in them. (See Demirdas et al, Ann Nutr Metab, 2017)
- Vitamine D in CGMP-AA is half of the amount in L-AA. Vit D deficiency is a general problem for those with and without PKU. Do you think this could affect the patients?
(See Demirdas et al, Ann Nutr Metab, 2017)
- Biotin is 1/6 in CGMP-AA when compared to L-AA.
- Line 127: correct spacing.
- Line 139 through line 156 --> a chronologic line (figure) explaining the study design would be very informative and much easier to understand.
Results
- Table 2 shows that the range for Tyr 10-200 in R3. Is this because the patient in question showed inability to adhere to their L-AA intake? If so, should this patient have been included in this study?
- Figures: please provide a legend for what the black lines mean. I assume this is the preferred range of Phe for age?
- Figures: why are you calling it GMP and not CGMP?
- Figure GMP minus Phe is not correctly displayed.
Discussion
- Line 266: spelling mistake hourf
- Line 269-271: how do you explain figure 3 showing both GCMP groups to have Phe ranges outside the recommended range for age?
- Line 281: please adjust this sentence, it is long and unclear.
- Line 294: the authors point out that fluctuation of Phe may affect IQ and also state that this hypothesis has not been sufficiently proven. What has been proven however, is that increased Phe has an adverse effect on IQ. Do they think that the fact that in the CGMP groups patient have higher Phe in several instances than advised by the Europea PKU guideline might have adverse effects? I find it alarming that in bot CGMP groups some patients (pediatric!) have high Phe.
- Please add your thoughts about the possible clinical implications directly for PKU patients. Do you think CGMP should be used more broadly? If so, what should be improved in order for patients to stay within advised Phe blood levels?
Author Response
Dear Reviewer
Thank you for taking the time to review this manuscript.
Please find the answers to your comments in the file attached.
Changes have also been made within the manuscript.
Thank you
